# Regulation of the Phosphoinositide Code by Phosphorylation of Membrane Readers

**DOI:** 10.3390/cells10051205

**Published:** 2021-05-14

**Authors:** Troy A. Kervin, Michael Overduin

**Affiliations:** Department of Biochemistry, University of Alberta, Edmonton, AB T6G 2H7, Canada; tkervin@ualberta.ca

**Keywords:** lipid specificity, membrane recognition, phosphoinositide binding, PX domain, protein phosphorylation, post-translational modification, regulation

## Abstract

The genetic code that dictates how nucleic acids are translated into proteins is well known, however, the code through which proteins recognize membranes remains mysterious. In eukaryotes, this code is mediated by hundreds of membrane readers that recognize unique phosphatidylinositol phosphates (PIPs), which demark organelles to initiate localized trafficking and signaling events. The only superfamily which specifically detects all seven PIPs are the Phox homology (PX) domains. Here, we reveal that throughout evolution, these readers are universally regulated by the phosphorylation of their PIP binding surfaces based on our analysis of existing and modelled protein structures and phosphoproteomic databases. These PIP-stops control the selective targeting of proteins to organelles and are shown to be key determinants of high-fidelity PIP recognition. The protein kinases responsible include prominent cancer targets, underscoring the critical role of regulated membrane readership.

## 1. Introduction

Membrane readers are protein domains that recognize the various PIPs found in each subcellular organelle and the plasma membrane. These conserved modules serve to reversibly attach proteins to lipid bilayers to mediate the assembly and disassembly of signaling and trafficking complexes. The best understood readers are the FYVE, PH and PX domain superfamilies, which represent the foundation of the phosphoinositide (PI) code that governs membrane recognition [1,2]. Of these, PX domains are uniquely dedicated and able to detect all PIP signals to mediate endosomal and plasma membrane trafficking of host proteins [3] as well as being engaged to traffic viral components including SARS CoV-2 proteins [4]. The PX superfamily is large, with 32,485 PX domains in the SMART database [5] including 8 distinct members in *Arabidopsis thaliana,* 15 in *Saccharomyces cerevisiae* and 49 in *Homo sapiens* (Figure 1). This superfamily is the focus here as it is particularly well characterized in terms of the structures and PIP specificities responsible for subcellular localization, and best embodies how the wider membrane code works.

Several mechanisms have been postulated to regulate membrane readers. A set of lipid kinases and phosphatases add and remove phosphates from inositol rings and are differentially localized [6]. However, this does not address how an individual membrane-localized protein can be selectively modulated without perturbing other proteins that recognize the same PIP ligand. Other partners in membranes could support selective interactions through coincidence detection [7]. However, this does not directly address how protein complexes assemble on membranes in the first place. The degradation of ubiquitinated proteins assembled on membranes occurs but can be slow [8]. Could another mechanism serve as a rapid, selective and direct way to control membrane readers?

Based on largescale analysis of existing data and modeling, we propose that the phosphorylation and dephosphorylation of PIP-specific binding surfaces provides custom control of individual membrane readers. These so-called PIP-stops [9] are not only conserved and widespread, as shown here, but they are concentrated in the sites of highest lipid specificity. We suggest that a variety of protein kinases and phosphatases are responsible for creating and removing PIP-stops from exposed lipid bilayer docking surfaces of membrane readers. Hence, we propose that this represents a dominant regulatory mechanism that is inherently responsive to signaling pathways and is disrupted by diseases including cancer. The ancient evolution and preservation of membrane code regulation suggests that it played a formative role in the organization of eukaryotic cells.

## 2. Materials and Methods

Protein sequences of all *Homo sapiens* and *S. cerevisiae* PX domains were selected from UniProt [10]. Alignments were generated with Clustal Omega [11] and manually adjusted using Jalview 2 [12] in order to refine the alignment of structural and functional elements. Mutational data were obtained from COSMIC [13], cBioPortal [14] and PhosphoSite [15].

Structures of all PX domains were obtained from the Protein Data Bank (PDB) [16] and the entry with highest resolution and best defined membrane binding elements was selected when choosing between multiple entries. Experimental structures of human PX domains that were assessed for the resolution of membrane binding sites included KIF16B: 2v14, 6ee0; NISCH: 3p0c; NOXO1B: 2l73; PIK3C2A: 2ar5, 2iwl, 2red, 2rea, 6bub; PIK3C2G: 2wwe; p40^phox^: 1h6h, 2dyb; p47^phox^: 1kq6, 1o7k, 1gd5; SGK3: 1xte, 1xtn, 6edx, 4oxw; SNX1: 2i4k; SNX3: 5f0j, 2yps, f50l, 2mxc, 5f0m, 5f0p; SNX5: 3hpc, 3hpb, 5tgi, 5tgh, 5tgj, 5wy2, 5tp1; SNX7: 3iq2; SNX9: 2raj, 2rak, 2rai, 3dyt, 3dyu; SNX10: 4on3, 4pzg; SNX11: 4ikb, 4ikd; SNX12: 2csk; SNX14: 4pqo, 4pqp, 4bgj; SNX15: 6ecm, 6mbi; SNX16: 5gw0, 5gw1; SNX17: 3lui, 3fog; SNX19: 4p2i, 4p2j; SNX22: 2ett; SNX24: 4az9; SNX25: 5woe, 5xdz; SNX27: 4has, 5zn9; SNX32: 6e8r and SNX33: 4akv. Those of yeast PX domains include Bem3: 6fsf; Grd19: 1ocs,1ocu; Mvp1: 6p0x; Vam7: 1kmd and Bem1: 2v6v, 2czo. If NMR or X-ray crystallographic structures or membrane binding site densities were lacking, the structure of the PX domain was constructed using I-TASSER, a threading-based protein structure prediction method [17]. The first model with the highest cluster size was selected as the representative structure. Structures were analyzed with PyMOL and ICM [18] in order to identify relationships between residues that are modified or membrane-interactive.

Membrane binding sites were predicted in each PX domain using the Membrane Optimal Docking Area (MODA) algorithm [19]. MODA assigns a score to each residue based on its likelihood of interacting with lipid bilayers, where several proximal scores greater than 30 indicate a highly probable membrane interaction site. MODA does not predict lipid specificity, but does map out likely membrane binding surfaces, including those that are novel, on protein structural models. MODA consistently predicted membrane interacting residues between the first and second β strands and stretching from the proline-rich element (PRE) to the second α helix, and more variably near the beginning of the first α helix, which is typically a little further from the membrane. This is consistent with experimental structures of PX domains complexed with lipids and micelles as well as mutational and binding studies (see below), and hence defines a consensus area for membrane docking. For ensembles of NMR structures, a representative model was selected for which the MODA scores for all residues in the β1-β2, β3-α1, and PRE-α2 sites were closest to the mean. Residues located in the consensus β1-β2, β3-α1 and PR-α2 elements (Sites 1, 2 and 3, respectively) with significant MODA scores, i.e., exceeding 30, were indicated on Figure 2. All phosphorylation sites are shown with the number of citations superscripted on the sequence alignment, and those within the boundaries of Sites 1, 2 or 3 were considered to be candidate PIP-stops.

The literature was surveyed for lipid binding data for each PX domain, with the relevant PIP specificities for human and yeast proteins shown in Table 1. Studies of PIP specificity and membrane affinity by PX domains of human proteins were examined including those of ARHGAP32 [3,20], ARHGAP33 [3,21], HS1BP3 [3,22], KIF16B [3,23,24], NISCH [3,25], NOXO1β [26,27,28,29], NOXO1γ [26,27,29], PIK3C2α [3,30,31], PIK3C2β [3,30], PLD1 [32,33,34], PLD2 [33,35,36,37], PXK [3,38], p40^phox^ [3,39,40,41], p47^phox^ [3,39,42], RPS6KC1 [3,43,44], SGK3 [3,45,46], SH3PXD2A [3,47], SH3PXD2B [47,48], SNX1 [3,49,50,51,52,53], SNX2 [3,49,52,54], SNX3 [3,9,49,51,55], SNX4 [3,56], SNX5 [3,57,58,59,60], SNX6 [3,61], SNX7 [3,55], SNX8 [62,63], SNX9 [3,64,65,66,67], SNX11 [3,68,69], SNX12 [3,51,70], SNX13 [3,71,72], SNX14 [3,71], SNX15 [3,73,74], SNX16, [3,51,75,76], SNX17 [3,77,78], SNX18 [79,80,81], SNX19 [3,71], SNX20 [82,83], SNX21 [82], SNX22 [3,84], SNX25 [3,71], SNX27 [3,85,86,87], SNX31 [3,88], SNX33 [89,90], PIK3C2γ, SNX10, SNX24, SNX29 and SNX32 [3] as well as yeast proteins Bem3, Grd19, Spo14, Ykr078w, Ypt35, Atg20, SNX41, SNX4, Vps17, Ypr097w [91], Bem1 [42,91,92], Mdm1 [91,93], Mvp1 [91,94], Vam7 [30,91,95,96,97] and Vps5 [30,91,96], with Ypr097W being a known outlier in sequence and function [91]. For clarity, NCF1 and NCF4 are popularly known as p47^phox^ and p40^phox^, respectively. In cases of conflicting PIP specificity data, quantitative data from liposomes and lipid-binding assays were given precedence over non-quantitative or non-bilayer assay data. Where there was conflicting quantitative experimental data on PIP binding specificity involving lipid bilayers, the protein’s subcellular localization was considered, and consistent scores were averaged.

Relative membrane specificities were quantified by introducing the Lipid Specificity Index (LSI). The PX domains that exhibit absolute specificity by significant recognition of only a single PIP were given an LSI of 10. Those PX domains which have no discernible affinity for PIPs were assigned an LSI of 0. PIP ligands were divided into two classes depending on whether they had 1 or over 1 terminal phosphates. Those PX domains with intermediate specificities were assigned an LSI based on the class of its predominant ligand with demerits for any additional ligands also bound. These PX domains were given an intermediate LSI value equal to 10 minus 1 for each additional in-class ligand and minus 2 for each additional out-of-class ligand. Hence, a PSI of 1 indicates a perfectly nonspecific domain that binds all seven PIPs. All integral values from 0 to 10 are theoretically possible, although there are fewer PX domains with moderate specificity scores.

A Membrane Affinity Index (MAI) was assigned to each PX domain based on whether its lipid bilayer interactions were strong (S), weak (W), or none (N), as listed in Table 1, based on the studies referenced above. The specific ligand of the PXDC1 PX domain is unknown, but divergences in its binding motifs including non-retention of all but one key basic position suggest that its ligand is very unlikely to be PI3P. While the lipid binding specificities of the PX domains of SNX30 and SNX33 have not yet been determined experimentally, they could be inferred. In particular, SNX30 is 68% and 100% identical to SNX7 in its PX domain and PIP binding sequences, while SNX33 is 77% and 100% identical to SNX18 in its PX and PIP binding sequences, respectively. Hence, these pairs of proteins, which also share similar modular architectures (Figure 1), are assumed here to have similar PIP specificities.

The PIP-Stop Score (PSS) was defined as a new parameter to estimate the likelihood that a given PX domain contains a PIP-stop that governs its binding to membranes. Post-translational modifications (PTMs) were obtained from cBioportal [14], dbPTM [98], PhosphoSite [15], qPTM [99] and an augmented human phosphoproteomic database [100], while yeast data were obtained from PhosphoGrid [101] and SuperPhos [102], *Drosophila melanogaster* data from iProteinDB [103] and *Danio rerio* and *Caenorhabditis elegans* data from PTMcode2 [104], providing multiple proteome-wide coverages, with our manual curation excluding duplicates. As a standardized measure that integrates data from multiple cell types, PSS balances out the effects of differential protein expression in various tissues, which can bias observed modification levels. We considered phosphorylation sites between the β1-β2 and PR-α2 elements and near the conserved YS sequence in α1 to be candidate PIP-stops due to their proximity to PIP binding sites and hence likely effects on modulating membrane interactions. After comparing the influence of various boundary functions based on different sequential and spatial distances from the center of the maximum MODA signal, we settled on a consensus boundary in the sequence alignment (Table 1) that spans PX membrane docking sites due to its concordance and accessibility.

Phosphorylation sites in any of the three sites mapped by MODA were added to generate the PIP-Stop Score (PSS) of each domain. Each residue that was experimentally determined to be phosphorylated by 1 cited study added 1 to the domain’s total score, those residues with 2–4 such citations added 2, and those with 5 or more citations added 3. The PSS of each human and yeast PX domain then indicates a standardized frequency of cellular control of its membrane binding surface. The protein kinases responsible for phosphorylating PIP-stops were predicted using NetworKIN and NetPhorest [105], with scores of at least 1 being considered significant, and where not attained, those of the highest scoring kinase was reported.

## 3. Results

### 3.1. Comparative Analysis of Membrane Binding Poses

The structures of 49 human PX domains were assembled to assess their membrane binding properties (Figure 2). This included 29 structures generated here using I-TASSER [17], all of which had acceptable qualities with average template modelling and confidence scores of 0.81 ± 0.09 and 0.38, while the folds and binding features of the remainder were determined previously by NMR and X-ray crystallography. Structural deviations were due to protein interaction features such as in the p47^phox^, phospholipase D (PLD), SNX5, SNX6 and SNX32 PX domains [106,107,108,109]. Nonetheless, there is clearly a conserved architecture that allows for the comparison of their binding and regulatory features.

Each structure was categorized by the PX domain’s Lipid Specificity Index in order to quantify its PIP selectivity (Table 1). We included a PIP as a ligand based on all the available experimental data and have focused only on PIPs, although other lipids such as phosphatidic acid can also stabilize bilayer complexes. Twenty PX domains recognized only one PIP and were assigned the maximum LSI value of 10. Three exhibited no significant membrane binding activity and thus had a lipid specificity of 0. Those with intermediate PIP specificities were assigned an LSI according to the aforementioned method. Together, this comprises the first essentially complete set of specificities and structures of a large family of membrane readers, providing a basis for examining how their binding and regulatory determinants relate.

The membrane binding sites of each PX domain structure were mapped using the membrane optimal docking area (MODA) algorithm. This identifies exposed residues that are likely to engage membrane surfaces based on analysis of a training set of membrane readers [19] (Figure 2). The consensus docking surface of PX domains comprises three proximal elements. Site 1 is a membrane insertion loop that connects the β1 and β2 strands. Site 2 is a canonical regulatory site first identified at the beginning of the a1 helix in several sorting nexins [9]. Site 3 spans a long irregular loop that in some structures encompasses a type II polyproline helix and can serve as an SH3 docking site [106]. This membrane interacting surface is consistent with PIP-bound structures of PX domains of Grd19, p40^phox^, p47^phox^, SNX3, SNX9 and SNX11 proteins [9,41,65,69,110,111,112], while SNX5, SNX6 and SNX32 bind proteins in the vicinity [109]. Thus, PX domains offer a common binding surface that recognizes the full spectrum of PIPs, with a few variations due to overlapping protein and lipid interaction sites.

### 3.2. Regulation Is Driven by PIP Specificity

Protein phosphorylation is thought to occur in intrinsically disordered regions and less frequently in protein:protein interfaces [113]. However, human PX domains exhibit 172 phosphosites which occur at frequencies similar to their entire protein sequences (that is, 2.9% versus 2.7% of residues, respectively). To investigate whether their membrane recognition surfaces are preferentially phosphorylated, we mapped all the available phosphoproteomic data. Of the 867 phosphorylation citations found in 37 human PX domains, 87.3% are found in Sites 1, 2 and 3, indicating preferential targeting of membrane binding interfaces. The introduction of phosphates at these sites would cause electrostatic repulsion of negatively charged phospholipid bilayers, and such events are known to delocalize sorting nexins from endosomal membranes and into the cytosol [9]. Hence, the membrane binding surfaces of most PX domains are subject to control by widespread phosphorylation, supporting the role of PIP-stops as important negative regulators of membrane targeting.

We investigated how the incidence of PIP-stop phosphorylation relates to the specificity of lipid recognition. Each structure was assigned a PIP-Stop Score to allow for the comparison of reported phosphorylations of membrane readers at any of their lipid binding sites. Plotting these PSS values against lipid specificities shows a dramatic skew, with the most frequent phosphorylation events being found mainly in the most selective domains (Figure 3). A similar skew is seen for each of the three component sites, suggesting that they are under similar pressure. This positive relationship suggests that all three sites contribute to ensuring high fidelity PIP recognition. A similar pattern is found in yeast (Figure 3d) suggesting an ancient imperative to regulate specific code readers, with the exception of one clear outlier in sequence, inferring a major responsibility in asserting precise subcellular targeting.

Several functional subsets can be identified. High numbers of citations for phosphorylation were found in sorting nexins 1, 2, 3, 12, 17 and 27 (Figure 4). These domains selectively recognize only PI3P or PI(3,4)P_2_ within endocytic pathways and are under the control of kinases that act on Site 2, while PLD2′s PX domain selectively recognizes PI(4,5)P_2_ on the plasma membrane subject to regulation by phosphorylation at all three sites. Many PX domains exhibit intermediate PIP specificities and phosphorylation frequencies. In contrast, the PX domains of ARHGAP33, SNX14 and SNX32 do not bind membranes and are rarely phosphorylated, with SNX32 being an outlier that is phosphorylated on an atypical helix–loop–helix insertion that mediates protein interactions [109]. Ten PX domains contain no known phosphosites yet bind PIPs, inferring less regulated or constitutive membrane interactions. For clarity, the Membrane Affinity Indices (MAIs) and PIP-Stop Scores of PX domains are not obviously correlated, indicating that kinase access is not significantly restricted by membrane occupancy. Rather, lipid specificity appears to be a primary predictor and likely impetus for this mode of membrane code regulation.

### 3.3. Kinases Acting on PIP-Stops

The presence of so many phosphoregulatory sites in membrane recognition sites implies that many protein kinases are involved. Human PX domains employ 50 pSer, 27 pThr and 29 pTyr residues in the diverse PIP-stops identified so far, indicating comparable levels of modification by Ser/Thr and Tyr kinases. Regulatory phosphorylations occur most frequently on the central serine of the superfamily’s most conserved motif inside Site 2, which is a predicted substrate for PKA, PKC and PAK kinases. However, 93 other PIP-stops are found on other membrane interacting elements, suggesting regulatory divergence. Comparing the patterns of Ser/Thr and Tyr phosphorylation reveals similarly skewed distributions towards highly specific membrane readers, indicating that the wider kinome is engaged to ensure PIP code control (Figure 3b,c). Nonetheless, the breadth of positions and sequences of PIP-stops suggests a multiplicity of the regulatory effectors, as well as potential for the tuning rather than total ablation of membrane binding.

The identities of the protein kinase and phosphatases partners that act on membrane readers are biologically key, with the levels of phosphorylation of PIP-stops being altered during cancer [114]. Many of the implicated kinases are membrane-localized and associate with membrane readers (Figure 5). For example, the EGFR and ErbB2 proteins interact with PIK3C2β and their kinase domains likely phosphorylate its pTyr-1401 PIP-stop [115]. Casein kinase Iα associates with SNX24 [116] and is predicted to phosphorylate its pThr-20 PIP-stop. The PIP-stops of p47^phox^ are substrates for kinases, including PAK1, through which it signals [117]. The PKCα protein associates with both the plasma membrane [118] and the SNX10 protein [119]. Several sorting nexins associate with activin receptors bearing Ser/Thr kinases [120] that are predicted to phosphorylate their PIP-stops. Jak kinases are found on membranes [121] where they can phosphorylate PIP-stops of several sorting nexins. Other enzymes including protein phosphatases also contribute, implicating their oncogenic functions. This infers the fidelity of PIP recognition is frequently deregulated in cancer, resulting in delocalization of many membrane readers and their partners.

Tool compounds are available to manipulate the regulators of membrane readers. The kinases that are most commonly identified as acting on PIP-stops are Jak1/2, PAK1, PKAα, PKCα and PKCβ (Figure 5), all of which are membrane-localized drug targets. Their involvement in cancer is well known, and inhibitors are available. PAK1 is localized to the cell periphery, where it contributes to gastric cancer progression [122] and mediates oncogenic signaling [123]. PKA is targeted to membrane compartments by A-kinase anchoring proteins [124] and is involved in tumor progression [125], while PKCs are critical enzymes for cancer intervention [126]. Targeting these kinases alters membrane recognition, with drug treatment perturbing the PIP-stops of several PX domains, as shown with Abl, BRAF-V600E and MKK1/2 kinase inhibitors [127,128]. Together, this emphasizes that understanding the membrane code is critical for navigating not only intracellular signaling but also the rational design of therapeutic agents.

Diagnostics are also impacted, with disease-linked mutations being located near PIP-stops. For example, mutations in p47^phox^ Site 2 including R42W/Q, E46K, E49K and of PIP-stop residue T48C are associated with chronic granulomatous disease and prostate, kidney, skin and intestinal cancer [13,129,130]. Mutations of Arg-90 in its Site 3 impair membrane localization and PIP binding [131] and are linked to intestinal, lung and skin cancers [132,133] and autoimmune diseases [134]. The R43W mutation beside SH3PXD2B’s PIP-stop causes Frank-Ter Haar syndrome [135] and is linked to colon cancer [13] while a proximal E96K mutation is linked to endometrioid carcinoma [13]. The SNX31 mutations T69I and D73H are proximal to its solitary PIP-stop at Thr-69 and are linked to skin and colorectal cancers [13,136]. Such mutations can now be predicted to alter the ability of membrane readers to dynamically and specifically recognize organelles and signal accurately, providing mechanisms for pathogenesis.

## 4. Discussion

Despite a billion years of evolutionary separation, some PIP-stops are absolutely conserved from yeast to humans. In particular, the RxYSxF sequence (where x is any residue) that constitutes the predominant PIP-stop in the PX superfamily (Figure 4) is present in 100% of the 248 complete SNX3 sequences available in Uniprot. The phosphorylation of this critical PIP-stop is apparent in yeast proteins Atg20, Grd19, SNX41 and Vam7, as well as fruit fly, zebrafish and nematode SNX1 and SNX3 proteins, indicating a pervasive regulatory significance in the development of endocytic traffic. Moreover, the skew of PIP-stop frequency towards the most specific PIP binders is also consistently observed (Figure 3d), indicating a longstanding pressure to control the most selective membrane readers.

We propose that PIP-stops regulate the events that lead to the maintenance, signaling and trafficking at most eukaryotic membrane-bound organelles. We have focused on PX domain-containing proteins here for practical reasons but suggest that a wider diversity of membrane readers could also be vulnerable to such regulatory modifications. Most PX domains are found on autophagosomes and early endosomes where their PI3P and PI(3,4)P_2_ interactions are most often toggled by PIP-stops in Site 2, as validated with sorting nexins 1, 3 and 12 [9]. Only SNX6 binds exclusively to PI4P in Golgi membranes [61] and is infrequently toggled by Site 3 PIP-stops, while no PX proteins appear to associate exclusively with late endosomal and lysosomal PI(3,5)P_2_ pools. Many PX proteins localize to the plasma membrane by less selective recognition of the abundant PI(4,5)P_2_ ligand as well as bursts of PI(3,4,5)P_2_ that are generated upon receptor stimulation, with medium levels of PIP-stop mediated control. As many of the PX:membrane interfaces are susceptible to modification at various positions, it is conceivable that the dynamics, selectivities and affinities of their various interactions could be tuned and adjusted by differentially localized protein kinases and phosphatases. Together, this constitutes a tantalizing type of control over the PI code that builds on the research of many groups and merits the investigation of its impact on cell biology and therapeutic intervention.

Extensions of the PI code from soluble membrane readers to transmembrane proteins and large assemblies can also be envisaged. For example, the divergent PIP specificities of membrane-spanning sorting nexins 13, 14 and 19 (Table 1) could serve to modulate their constitutively localized signaling functions. Multiprotein complexes can be assembled by PX domains that recognize PIPs while binding retromers [137] or pathogen proteins [109], with additional layers of regulation provided via phosphorylation events that alter protein–protein interactions [131]. The revelation that a preponderance of PIP-stops may exist to control much of the PX superfamily suggests broader applicability across the proteome. A huge range of protein domains recognize PIPs, as well as a variety of other phospholipids, sphingolipids and glycolipids. In principle they could also exhibit similar regulatory mechanisms, as could the adjacent modules that support membrane readers through coincidence detection of lipids. The development of technologies such as styrene maleic acid lipid particles (SMALPs) to isolate and resolve native membrane:protein complexes in nanodiscs could illuminate more PIP targets [138], and other PTMs could also be exerting regulatory effects. While the diverse folds, sites, modifications and specificities involved necessitate further analysis, the extension of the membrane code will be enabled by the tools and principles presented here.

## Figures and Tables

**Figure 1 cells-10-01205-f001:**
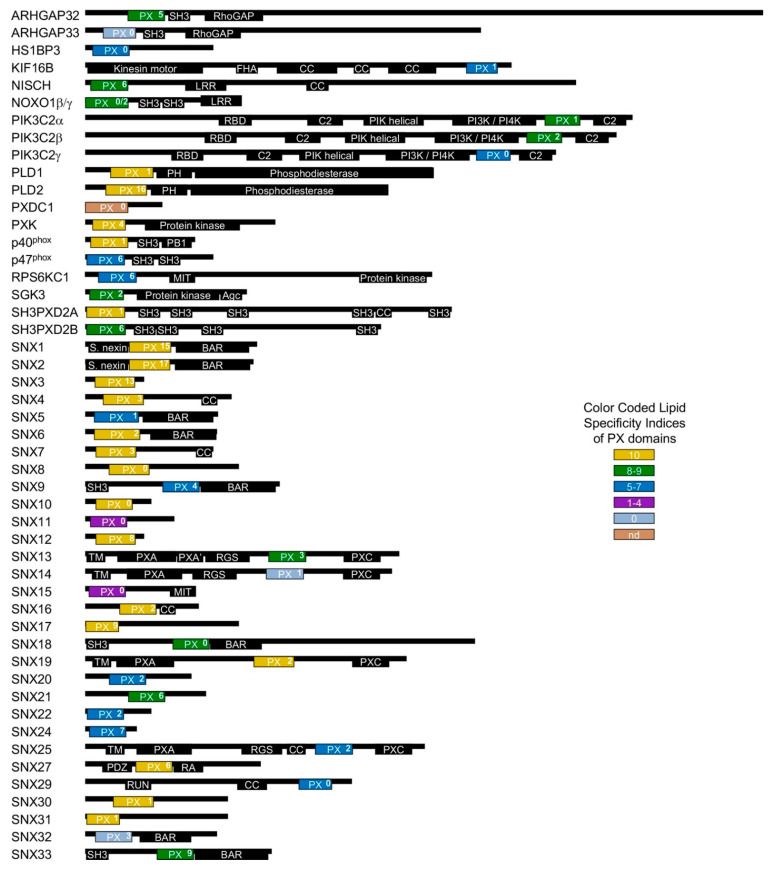
Human PX domain-containing proteins. PX domains are shown as boxes colored by specificities as in Figure 2 with PSS values superscripted. Other domains are black and labelled as Agc (AGC kinase C-terminal), BAR (Bin–Amphiphysin–Rvs), C2 (protein kinase C conserved region 2), CC (coiled coil), GAP (GTPase-activating protein), FHA (forkhead-associated), LRR (leucine-rich repeat), MIT (microtubule interacting and transport), PB1 (Phox and Bem1), PDZ postsynaptic density 95, disk large, zonula occludens), PH (pleckstrin homology), PIK (phosphoinositide 3-kinase), PXA (PX-associated), PXC (PX C-terminal), RA (Ras-associating), RBD (Ras binding domain), RGS (regulator of G protein signaling), RUN (RPIP8, unc-14 and NESCA), SH3 (src homology 3), or TM (transmembrane).

**Figure 2 cells-10-01205-f002:**
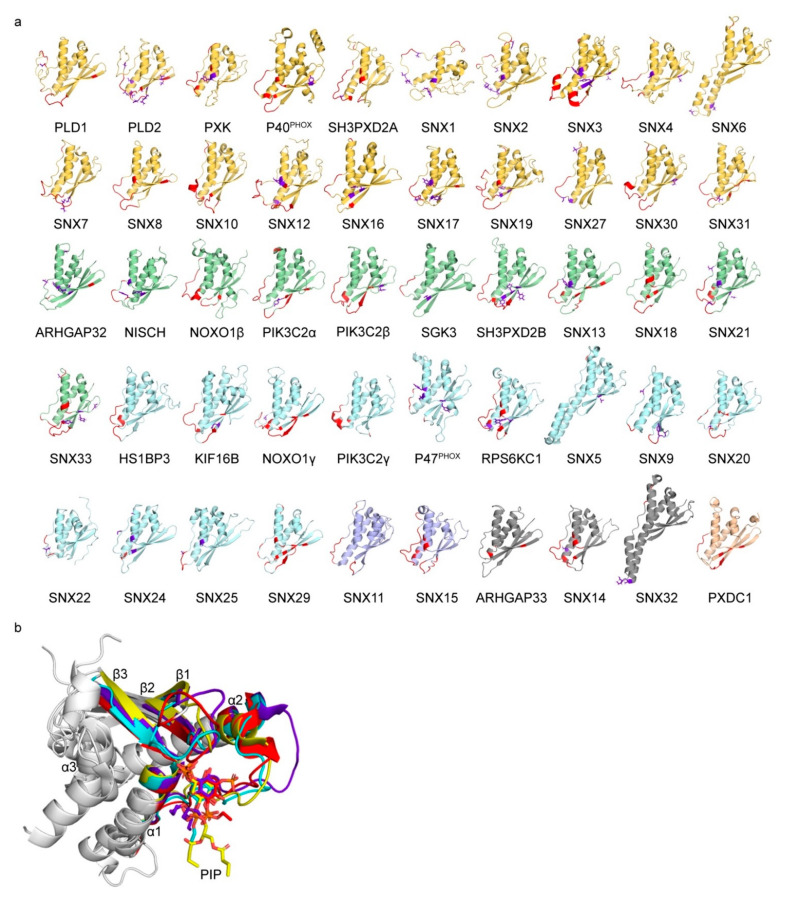
Structures of all human PX domains. (**a**) Ribbon models are shown with residues in membrane-interacting sites based on MODA scores greater than 30 and are shown in red. Phosphorylated residues in Sites 1, 2, and 3 are shown in purple with their side chains indicated. Backbone ribbon colors correspond to LSI values of 1–4 (purple); 5–7 (light blue), 8–9 (green), 10 (yellow), does not bind PIPs (grey) and no data available (beige). (**b**) The structural elements of superimposed PIP-bound PX domain structures of Grd19 (1ocu), p40^phox^ (1h6h), SNX9 (2rak), and SNX11 (6koj) are labelled with Sites 1, 2, and 3, and the carbon atoms of the PIPs belonging to each structure are colored red, yellow, cyan and purple, respectively.

**Figure 3 cells-10-01205-f003:**
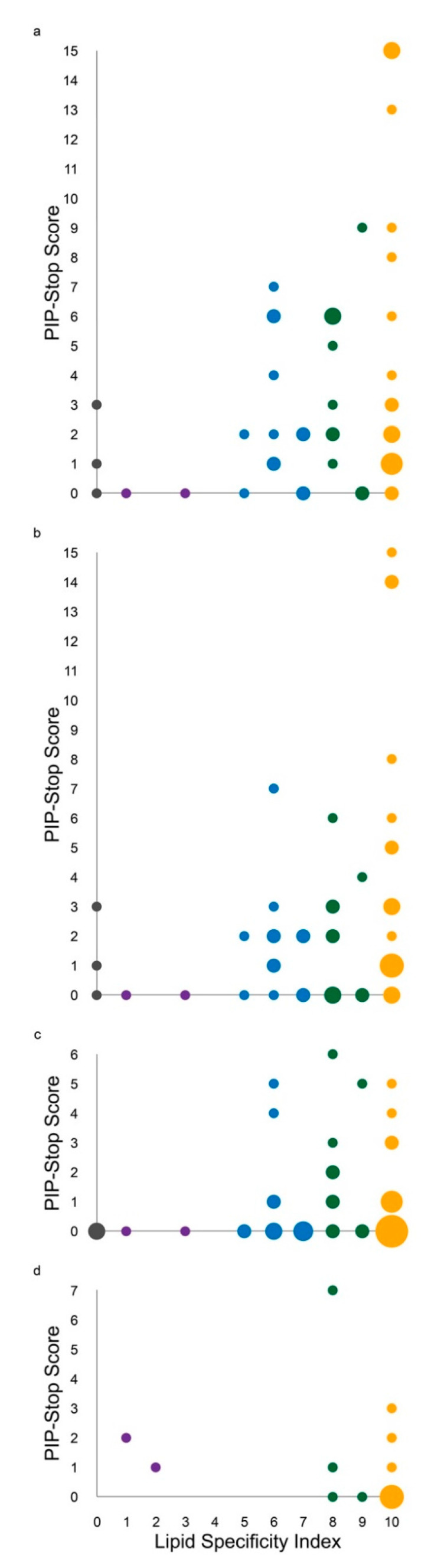
Relationship between ligand specificity and phosphorylation of membrane readers. The PSS values for (**a**) all, (**b**) pSer/pThr and (**c**) pTyr residues in human and (**d**) *S. cerevisiae* PX domains are plotted against LSI values (and capped here at 15) for each PX domain, with circle areas being proportional to the number of proteins occupying a position and being colored as in Figure 2.

**Figure 4 cells-10-01205-f004:**
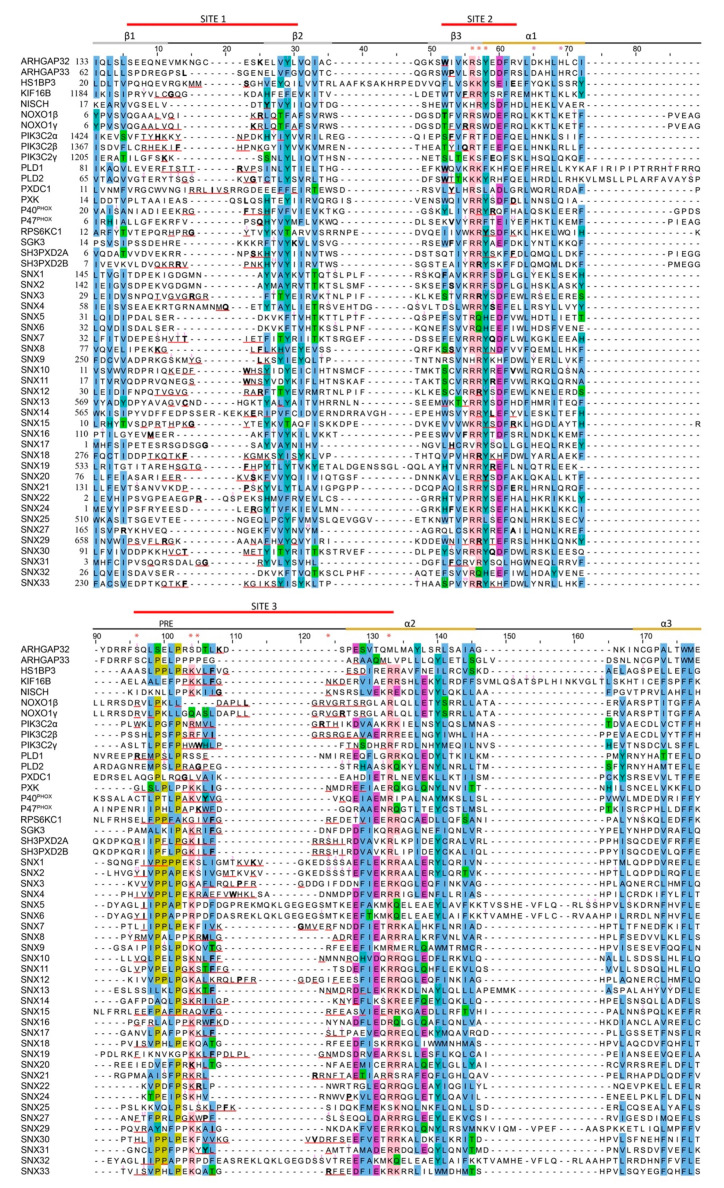
Alignment of the sequences of human PX domains. Phosphorylation sites are annotated with a superscript indicating the number of citations (capped at 99). The 3 membrane binding Sites are indicated with red lines, as are the residues in each Site with MODA scores exceeding 30, and the residue with the highest MODA score in each Site is bolded. Conserved residues are highlighted blue (hydrophobic), olive (Pro), green (polar), acidic (purple), and pink (basic). The secondary structure and PRE are labelled above the sequences. PIP and phosphatidic acid binding residues are asterisked in red and purple.

**Figure 5 cells-10-01205-f005:**
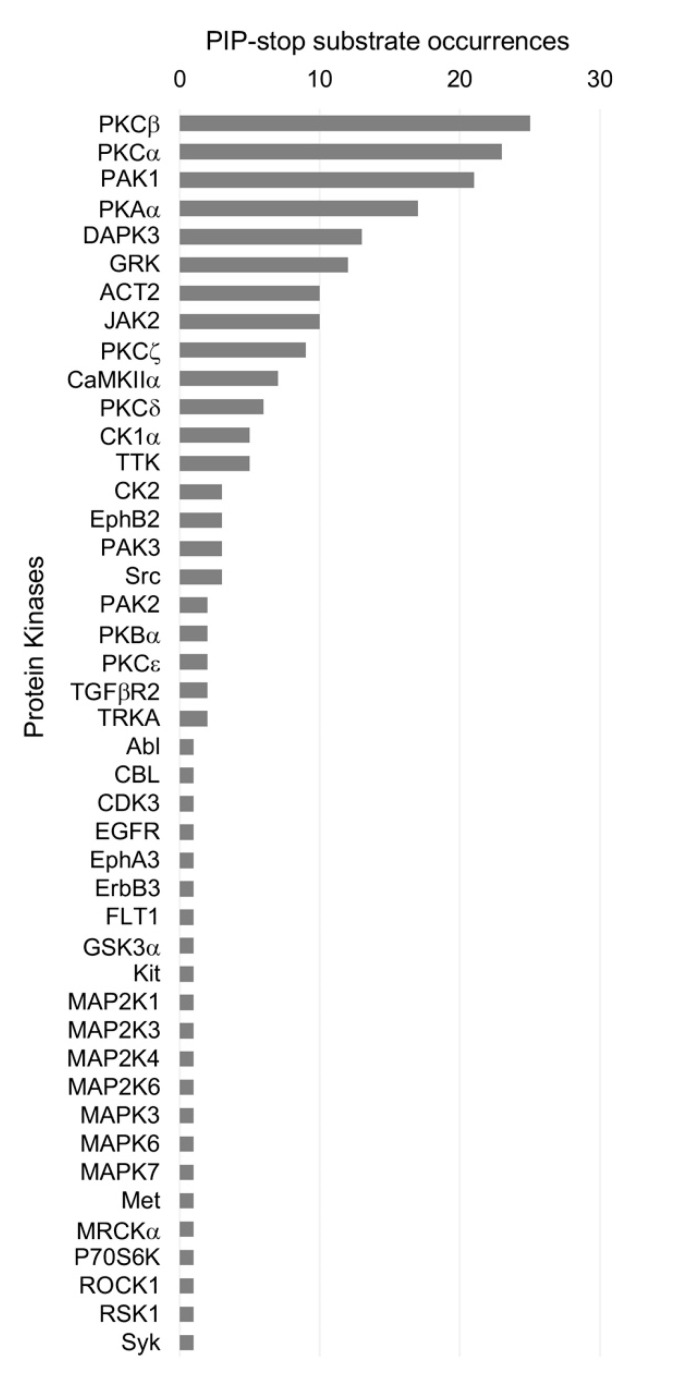
Protein kinases that act on PIP-stop motifs. The total number of occurrences of each human kinase which is predicted by NetworKIN and NetPhorest [105] to phosphorylate the candidate PIP-stops identified here in all human PX domains are shown.

**Table 1 cells-10-01205-t001:** Properties of PX domains (below).

***H.s. Protein***	**LSI**		**PIP Ligands**		**PSS**		**MAI**		**PDB**
PLD1	10		345		1		S		IT
PLD2	10		45		16		W		IT
PXK	10		3		4		W		IT
p40^phox^	10		3		1		S		1h6h
SH3PXD2A	10		3		1		S		IT
SNX1	10		34		15		S		2i4k
SNX2	10		34		17		S		IT
SNX3	10		3		13		S		5f0j
SNX4	10		3		3		W		IT
SNX6	10		4		2		W		IT
SNX7	10		3		3		W		IT
SNX8	10		3		0		S		IT
SNX10	10		3		0		W		4on3
SNX12	10		3		8		S		2csk
SNX16	10		3		2		S		5gw0
SNX17	10		3		9		S		IT
SNX19	10		3		2		S		IT
SNX27	10		3		6		S		4has
SNX30	10		3		1		W		IT
SNX31	10		3		1		S		IT
NOXO1β	9		45,345		0		W		2l73
SNX18	9		34,45		0		S		IT
SNX33	9		34,45		9		S		IT
ARHGAP32	8		3,4,5		5		W		IT
NISCH	8		3,34		6		W		3p0c
PIK3C2α	8		34,35,45		1		S		2ar5
PIK3C2β	8		34,45,345		2		S		IT
SGK3	8		3,34		2		S		1xte
SH3PXD2B	8		3,34		6		S		IT
SNX13	8		3,34		3		W		IT
SNX21	8		3,45		6		S		IT
NOXO1γ	7		4,5,35		2		W		IT
PIK3C2γ	7		34,35,45,345		0		W		2wwe
SNX25	7		34,35,45,345		2		S		5woe
SNX29	7		3,34,45		0		S		IT
KIF16B	6		3,34,45,345		1		S		2v14
p47^phox^	6		3,34,45,345		6		S		1kq6
RPS6KC1	6		3,34,45,345		6		S		IT
SNX5	6		3,34,35,45		1		W		3hpc
SNX9	6		3,34,45,345		4		W		2raj
SNX22	6		3,34,45,345		2		S		2ett
SNX24	6		3,34,35,45		7		S		4az9
HS1BP3	5		3,34,35,45,345		0		S		IT
SNX20	5		3,5,35,45		2		S		IT
SNX15	3		3,4,34,35,45,345		0		S		IT
SNX11	1		3,4,5,34,35,45,345		0		S		4ikb
ARHGAP33	0		0		0		N		IT
SNX14	0		0		1		N		IT
SNX32	0		0		3		N		6e8r
PXDC1	nd		nd		0		n.d.		IT

***S.c. Protein***		**LSI**		**PIP Ligands**		**PSS**		**PDB**	
Bem1		10		4		0		2v6v,2czo	
Bem3		10		3		2		6fsf	
Grd19		10		3		3		1ocs,1ocu	
Mdm1		10		3		0			
Mvp1		10		3		0		6p0x	
Spo14		10		3		0			
Vam7		10		3		1		1kmd	
Ykr078w		10		3		0			
Ypt35		10		3		0			
Vps5		9		3,5		0			
Atg20		8		3,4,5		1			
Snx4		8		3,45		0			
Snx41		8		3,4,5		7			
Vps17		2		3,4,5,34,35,45		1			
Ypr097w		1		3,4,5,34,35,45,345		2			

Those of *H. sapiens* and *S. cerevisiae* proteins and their LSI, PSS and MAI values are listed, with MAIs being strong (S), weak (W) or none (N). Consensus PIP ligands are denoted by phosphate positions, e.g., PtdIns(3,4,5)P3 is denoted “345”, while “nd” indicates not determined. PDB entries used here are listed, with I-TASSER-derived structures denoted “IT”. Box colors correspond to LSI values of 10 (yellow), 8–9 (green), 5–7 (aqua), 1–4 (purple), 0 (grey) or nd (beige).

## Data Availability

The databases and software used here are publicly available.

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
