# Peer review of "Regulation of the Phosphoinositide Code by Phosphorylation of Membrane Readers"

_cells, 2021, doi:10.3390/cells10051205_

Round 1
Reviewer 1 Report
This is a very nice article that propose a general model for the regulation of phospholipid binding by PX domains. The model proposed is interesting and, certainly, will have significant impact in the field if subsequently demonstrated empirically. The manuscript is very well written, with good description of the methodology and results obtained. I have no reservations about its quality. Minor issues are:
- Title. I would indicate in the title that this is a model. The manuscript lacks experimental demonstration that this model does take place in nature and that is as general as it seems.
- I would place Table 1 before Figure 2.
- Figures 1 and 2. Legend at the bottom, not at the top.
- Figure 3. Legend at either the bottom or top, not sandwiched between two paragraphs.
Author Response
This is a very nice article that propose a general model for the regulation of phospholipid binding by PX domains. The model proposed is interesting and, certainly, will have significant impact in the field if subsequently demonstrated empirically. The manuscript is very well written, with good description of the methodology and results obtained. I have no reservations about its quality. Minor issues are:
- Title. I would indicate in the title that this is a model. The manuscript lacks experimental demonstration that this model does take place in nature and that is as general as it seems.
Authors Response: We thank the reviewer for this comment as well as the related comment by the second reviewer and have changed the title to be more descriptive, i.e: “Regulation of the phosphoinositide code by phosphorylation of membrane readers” and emphasize in the abstract that we are analyzing existing and modelled structures.
- I would place Table 1 before Figure 2.
Authors Response: We have made the requested change.
- Figures 1 and 2. Legend at the bottom, not at the top.
Authors Response: We have made the requested change.
- Figure 3. Legend at either the bottom or top, not sandwiched between two paragraphs.
Authors Response: We have put the legend at the bottom as requested.
Reviewer 2 Report
In their manuscript, Kervin & Overdui establish phosphatidylinositol phosphates (PIPs) as a general membrane code, which attracts Phox homology (PX) domain-containing proteins. These domains are largely present over various taxa and bind PIPs with broad specificity, and as such perform e.g. signaling roles. A substantial amount of structural information exists on PX domains from many species, including humans. The authors use this information, together with earlier interactome data and computational models of experimentally uncharacterized PX domains, to conclude how phosphorylation and dephosphorylation of PX domains could regulate PIP interactions and membrane binding. The authors score members of the domain family using computational methods and provide rationale into how PX domain-containing proteins are regulated in regard to PIP binding and recognition. The authors divide the known human PX-domains into clades based on PIP-binding specificities, domain structures and membrane reader site phosphorylation, establishing that phosphorylation of PX domains at specific sites is a major governing force of PIP recognition and binding.
The manuscript is certainly interesting and well written. I especially appreciate the effort the authors took in explaining the method part of the study. All in all, the manuscript caters a lot of novel ideas on the regulation of PX domain binding to membranes, but there are certain aspect of the study that should be addressed for the sake of clarity when considering the nature and methodology of the study. Therefore, I kindly ask the authors to revise the manuscript based on the following points that I have gathered:
- Title – the authors claim the study to be “The structural basis of the regulation of the membrane code”.
- However, the study seems to be specifically about PIP recognition, binding, and the involved regulation. The authors even mention this on lines 173 – 174. Membrane recognition in biological systems also involves other lipids than PIPs, such as various acidic lipids, cholesterol, sphingolipids and gangliosides. Therefore, I ask the authors to modify the title to better describe the study, namely, the specific focus on PIPs over other membrane components.
- Additionally, while the authors discuss several known structures of PX domains and many more that are modeled using I-TASSER, the title implies that the study is about the structural basis of the regulation, although what the authors present is more the molecular basis. To do the former, the authors should compare the available structural data and the modeled structures more thoroughly with one another through superpositions as well as by adding phosphoresidues in selected membrane reading sites and performing MD simulations to visualize whether PIP binding is affected or not. If this is not possible, the title wording should be revised.
- Abstract – the authors should clearly state that the study relies solely on modeling, bioinformatics and earlier structural data.
- Introduction – on lines 39 – 41, the authors propose that different phosphorylation states control the function on membrane readers. However, it would be welcome to state that the proposal is based on a large-scale analysis of existing data, supplemented by modeled structures, rather than wet experiments that specifically aim to demonstrate this. Additionally, do the authors have evidence for the control being dominant and rapid, as stated here? If so, past literature should be cited or novel experimental data should be presented.
- Introduction/discussion – could the authors perhaps briefly write about the relative abundances of different PIPs with each other and other lipid types in different membranes or cell compartments? How does this correlate with the presence of PX domain proteins in regards to their subcellular localization? Does the PIP membrane code change between different membranes and do the reading proteins follow any trends? Whilst this does not directly relate to the regulation of membrane code, it does relate to the membrane code itself that the authors have now established as a term.
- Starting from line 201 (The consensus docking surface…) – I feel that the inclusion of a figure that illustrates the discussed secondary structure features for Sites 1 – 3 would greatly aid the readers of the manuscript. It would also help to superpose a few selected PX domain structures and highlight any key differences in the structures that encompass the three Sites, if there are any differences. Looking at Fig. 2, while many of the proteins seem similar in folding, they are certainly not identical.
- Given the large amount of experimental and modeled structural PX domain data, can the authors explain the structural determinants that govern the (un)specificity of PIP binding? Can the data be used to predict the ligand of PXDC1, for which PIP specificity data is unavailable?
- Minor points
- Abstract: On line 12, the authors write “These PIP stops control selective organelle localization” – what do the authors mean by this? Localization of PX domains to different organelles? Kindly rephrase for clarity.
- On line 58, you can abbreviate Homo as H., since Homo sapiens has been mentioned earlier in Materials and Methods.
- In Fig. 2, the colors magenta and red are too similar at least when the manuscript is printed on paper. Consider changing one of them to stand out better.
- In Fig. 3, the image quality is compromised by compression artifacts, at least in the version I received – please upload a higher quality figure if possible.
- Line 260: “…, indicating comparable …”; the space in between the two words might be a double space, please check.
Author Response
The manuscript is certainly interesting and well written. I especially appreciate the effort the authors took in explaining the method part of the study. All in all, the manuscript caters a lot of novel ideas on the regulation of PX domain binding to membranes, but there are certain aspect of the study that should be addressed for the sake of clarity when considering the nature and methodology of the study. Therefore, I kindly ask the authors to revise the manuscript based on the following points that I have gathered:
- Title – the authors claim the study to be “The structural basis of the regulation of the membrane code”.
- However, the study seems to be specifically about PIP recognition, binding, and the involved regulation. The authors even mention this on lines 173 – 174. Membrane recognition in biological systems also involves other lipids than PIPs, such as various acidic lipids, cholesterol, sphingolipids and gangliosides. Therefore, I ask the authors to modify the title to better describe the study, namely, the specific focus on PIPs over other membrane components.
Authors Response: We thank the reviewer for this comment as well as the related comment by the first reviewer and have changed the title to be more descriptive, i.e.: “Regulation of the phosphoinositide code by phosphorylation of membrane readers”. We also mention the importance of other lipids in the new statement in the concluding paragraph: “Other domains that recognize PIPs as well as other phospholipids, acidic lipids, choles-terol, sphingolipids and gangliosides could exhibit similar regulatory mechanisms…”
- Additionally, while the authors discuss several known structures of PX domains and many more that are modeled using I-TASSER, the title implies that the study is about the structural basis of the regulation, although what the authors present is more the molecular basis. To do the former, the authors should compare the available structural data and the modeled structures more thoroughly with one another through superpositions as well as by adding phosphoresidues in selected membrane reading sites and performing MD simulations to visualize whether PIP binding is affected or not. If this is not possible, the title wording should be revised.
Authors Response: We have changed the title as suggested,altered the abstract to state that we are anlyzing existing and modelled structures, and have included a new figure panel (2b) showing the superimposed structures of PX:PIP complexes.
- Abstract – the authors should clearly state that the study relies solely on modeling, bioinformatics and earlier structural data.
Authors Response: We have made the requested change by stating in the revised abstract that: “Here we reveal that throughout evolution these readers are universally regulated by phosphorylation of their PIP binding surfaces based on our analysis of both existing and modelled structures and phosphosphoproteomic databases”.
- Introduction – on lines 39 – 41, the authors propose that different phosphorylation states control the function on membrane readers. However, it would be welcome to state that the proposal is based on a large-scale analysis of existing data, supplemented by modeled structures, rather than wet experiments that specifically aim to demonstrate this.
Authors Response: We have made the requested change to the text: “Based on largescale analysis of existing data and modeling we propose that phosphorylation and dephosphorylation of PIP-specific binding surfaces is the most dominant, rapid and dynamic way for custom control of individual membrane readers.”
- Additionally, do the authors have evidence for the control being dominant and rapid, as stated here? If so, past literature should be cited or novel experimental data should be presented.
Authors Response: We have clarified this issue by stating that: “We suggest that a variety of protein kinases and phosphatases are responsible for creating and removing PIPstops from exposed lipid bilayer docking surfaces of membrane readers. Hence we propose that this represents a dominant regulatory mechanism that is inherently responsive to signaling pathways.”
- Introduction/discussion – could the authors perhaps briefly write about the relative abundances of different PIPs with each other and other lipid types in different membranes or cell compartments? How does this correlate with the presence of PX domain proteins in regards to their subcellular localization? Does the PIP membrane code change between different membranes and do the reading proteins follow any trends? Whilst this does not directly relate to the regulation of membrane code, it does relate to the membrane code itself that the authors have now established as a term.
Authors Response: This is an excellent point, and we have added a paragraph to address this issue in the discussion.
- Starting from line 201 (The consensus docking surface…) – I feel that the inclusion of a figure that illustrates the discussed secondary structure features for Sites 1 – 3 would greatly aid the readers of the manuscript. It would also help to superpose a few selected PX domain structures and highlight any key differences in the structures that encompass the three Sites, if there are any differences. Looking at Fig. 2, while many of the proteins seem similar in folding, they are certainly not identical.
Authors Response: We have added the requested figure panel (2b) showing several illustrative PIP binding sites with secondary structure features in greater detail.
- Given the large amount of experimental and modeled structural PX domain data, can the authors explain the structural determinants that govern the (un)specificity of PIP binding? Can the data be used to predict the ligand of PXDC1, for which PIP specificity data is unavailable?
Authors Response: This is a great point and we have added this statement: “The specific ligand of the PXDC1 PX domain is unknown, but divergences in its binding motifs including non-retention of all but one key basic position suggest that its ligand is unlikely to be PI3P.”
- Minor points
- Abstract: On line 12, the authors write “These PIP stops control selective organelle localization” – what do the authors mean by this? Localization of PX domains to different organelles? Kindly rephrase for clarity.
- On line 58, you can abbreviate Homo as H., since Homo sapiens has been mentioned earlier in Materials and Methods.
- In Fig. 2, the colors magenta and red are too similar at least when the manuscript is printed on paper. Consider changing one of them to stand out better.
- In Fig. 3, the image quality is compromised by compression artifacts, at least in the version I received – please upload a higher quality figure if possible.
- Line 260: “…, indicating comparable …”; the space in between the two words might be a double space, please check.
Authors Response: We have rephrased the text and figures as suggested.